# Civil Society and Social Integration of Asylum Seekers: The 'Strength of Weak Ties' and the Dynamics of 'Strategic Action Fields'

Lennart Olsson [1,*] , Anne Jerneck [1], Claudia Fry [2] and Anika Binte Habib [1]

1   Centre for Sustainability Studies (LUCSUS), Lund University, SE-221 00 Lund, Sweden;
    anne.jerneck@lucsus.lu.se (A.J.); anika_binte.habib@lucsus.lu.se (A.B.H.)
2   Environment and Migration: Interactions and Choices Section (EMIC), United Nations University,
    D-53113 Bonn, Germany; fry@ehs.unu.edu
*   Correspondence: lennart.olsson@lucsus.lu.se

**Abstract:** The dramatic increase of asylum seekers entering the EU in 2015 profoundly changed migration politics in many EU countries. Not least in Sweden which accepted more migrants per capita than any other EU country but then swung abruptly to become among the strictest recipient countries. We use Sweden as a critical and extreme case to argue that the rapid shift in asylum politics and public opinion towards migration is not profoundly shared in society. Based on a local media analysis of three types of purposively selected rural municipalities followed by the analysis of a survey of civil society organizations sent to all 290 municipalities in Sweden, we find strong civil society support and willingness to both receive migrants and facilitate their integration into society. Despite increasing votes for political parties with anti-migration policies, we also find remarkably positive attitudes towards migrants in civil society organizations and among citizens in the EU27 barometer for Sweden. The upshot is optimism that civil society can balance the anti-immigration governance imposed by both left and right political regimes and that populism will succeed only if it has the capacity to thoroughly transform civil society attitudes—which we doubt.

**Keywords:** civil society; integration; migration; populism; rural; Sweden

## 1. Introduction

This article aims to better understand the real and potential role of civil society in assisting asylum seekers and facilitating their integration into the host country, taking Sweden as a critical and extreme case. Existing research on migration and integration in Europe is centered on major population centers, while we focus more on the role of civil society in supporting the reception and facilitating the integration of asylum seekers in small and medium-sized rural municipalities. Further, we note the tension in opinions between anti-migration sentiments expressed in the electorate in response to public opinion and political party rhetoric versus the positive attitudes expressed in the EU 27 barometer for Sweden and in our survey. Hence, we ask: how can this discrepancy be explained, and to what extent can civil society counteract anti-migration attitudes?

The selection of Sweden is of strategic importance for two main reasons. It is a critical case for understanding a sudden policy shift after the 2015 in-migration, and it is an extreme case as regards the discrepancy in opinion between populist antimigration sentiments in the electorate and the positive attitude towards migrants in civil society. A critical case is either the *most* or the *least* likely to fulfill a certain proposition (Flyvbjerg 2006). Given Sweden's long history of in- and out-migration in an ever more established and well-functioning welfare state along with its generous and, at times, very active migration policy, it was 'most likely' that Sweden would welcome asylum seekers in 2015. But it was 'least likely' that Sweden would suddenly shift to a very restrictive migration policy in the wake of the

'European Asylum Crises'. Sweden is also an extreme case. Given the positive attitudes in Sweden towards migration both in the EU 27 barometer and in civil society playing a very active and central role in the 2015 reception, and in the further integration of asylum seekers after that, it is a surprising contrast that a populist party with an explicit antimigration agenda receives an increasing share of political votes. Taken together, these conditions make Sweden an extreme case that may reveal compelling findings about complex dynamics.

We have settled for an open definition of integration as 'the process of becoming an accepted part of society' (Penninx and Garcés-Mascareñas 2016). That process raises further questions on how to conceptualize and implement rights, obligations, and reciprocity; how to explain the dynamics between economic and social integration; and how to include new actors and perspectives. Our focus is explicitly on civil society and the integration of asylum seekers/grantees and not on the migration of cheap labor often driven by 'the need *for* rather than the needs *of* low-wage migrant workers' (Rye and Scott 2018).

### 1.1. Migration and Demography

In 2015, the UNHCR recorded the highest number of displaced people (65.3 million) since its establishment in 1949. Of those, one-third were refugees (21.3 million). Although questioned by scholars (Schenner and Neergaard 2003; Stiernström et al. 2019; Fry and Islar 2021), the rapid influx to Europe of over one million asylum seekers in 2015 is often labeled the 'European crisis of migration' (Almustafa 2021). Within a few months in late 2015, Sweden received more asylum seekers per capita than any other country in the EU (163,000) and well beyond what Sweden received either before or after that (Figure 1).

In response, the number of civil society organizations and networks quadrupled and were praised for their capacity to quickly engage, collaborate, coordinate activities, and start solving problems. At least every other municipality depended on the readiness of civil society organizations to assist asylum seekers on arrival and help them access accommodation and facilities such as home schooling, language cafés, legal assistance, translation work, trauma treatment, internships, and leisure activities (SOU 2017).

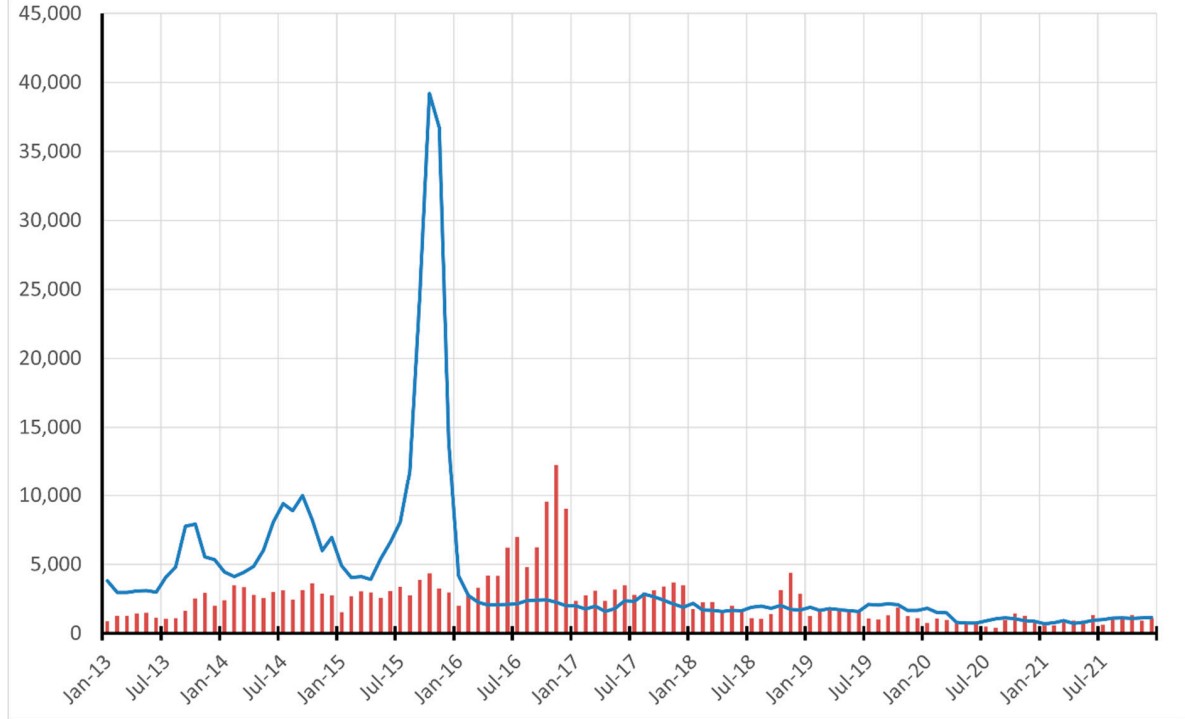

**Figure 1.** Number of asylum applications (line) and approved applications (column) per month in Sweden 2013–2021. Source of data: Statistics Sweden (applications) and Swedish Migration Agency (approved applications).

Historically, Sweden has experienced substantial migration flows. When the massive out-migration of farming smallholders in 1850–1930 turned into an in-migration of industrial labor in 1950–1970, the share of foreign-born people tripled to 7 percent in two decades (Hjern 2012). Again, from 2000 to 2021, the share of people born outside of the EU increased, now mainly through asylum immigration, from 6 to 15 percent, the highest percentage share in the EU (Örstadius 2022).

Geographically, the population in Sweden is unevenly distributed, with only 10 percent in the whole of the north covering 60 percent of the country versus 25 percent in the three largest cities—all located in the south (SCB 2023b). Rapid rural out-migration from the north implies that small municipalities struggle to maintain basic welfare, social services, and job opportunities (Hedberg and Haandrikman 2014) while small dynamic municipalities in the south struggle to attract skilled labor (Arbetsförmedlingen 2022). This has resulted in a complicated situation where political parties compete in promising services and favorable labor policies for rural development while at the same time advocating increasing restrictions on asylum migration to Sweden.

In 2013–2023, migration politics and the number of asylum seekers in Sweden varied like a rollercoaster to peak in 2015 (Figure 1). After that, Sweden introduced firmer border control, time-limited residence permits, and reduced options for family reunification, and thus settled at the minimum level required by the Court of Justice and international conventions (Righard and Öberg 2019). More recently, the government that came into power in the autumn of 2022 has further restricted the possibility for people to seek asylum in Sweden, intending to have the EU's most restrictive migration policy. This includes a reduction of quota refugees from 5000 per year (since 2018) to 900 per year (Migrationsverket 2022).

*1.2. Migration and Attitudes—In Sweden and in the EU*

According to the special Eurobarometer on the Integration of non-EU immigrants in the EU, the attitude towards non-EU migrants is much more positive in Sweden than in the rest of the EU27 (Figure 2). People in Sweden also state that they interact more often with non-EU migrants than the average in the EU27 (Figure 3). The survey defines immigrants as people born outside the EU who have moved away from their country of birth and are currently staying legally in an EU country.

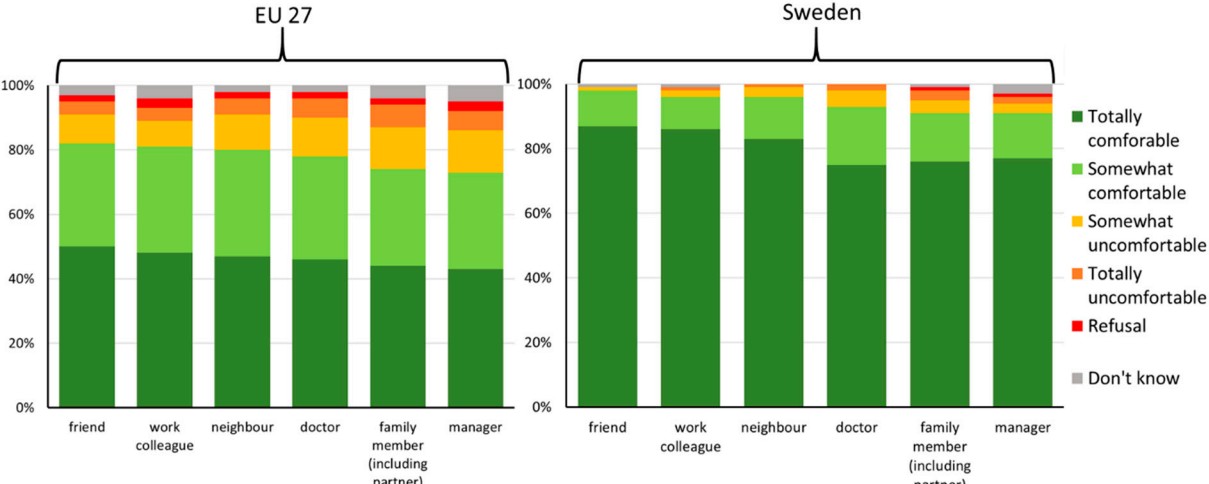

**Figure 2.** Attitudes towards non-EU migrants in EU27 (**left**) and Sweden (**right**). Responses to the question: Would you personally feel comfortable or uncomfortable having an immigrant as your friend/work colleague/neighbor/doctor/family member/manager? The interviews (face-to-face) were carried out in the period 2–29 November 2021. The EU27 data contain 26,510 interviews, and the Swedish data contain 1070 interviews.

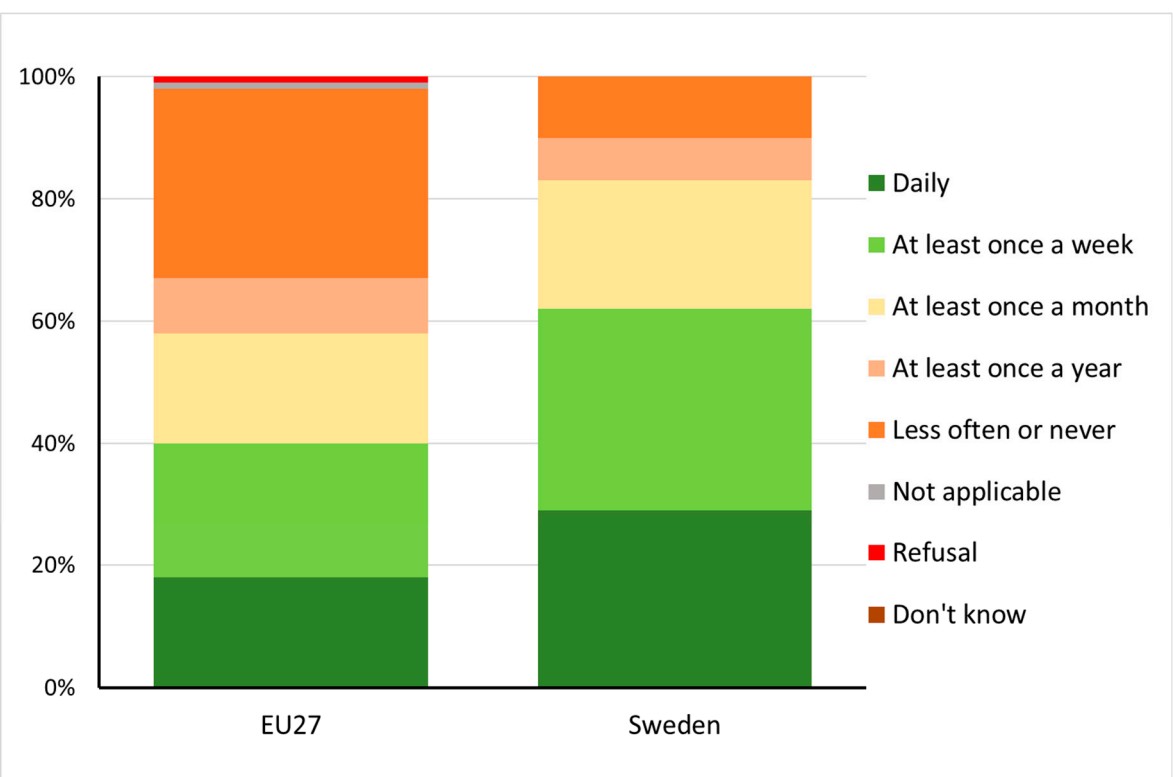

**Figure 3.** Interaction with non-EU migrants in daily life in the EU27 (**left**) and Sweden (**right**). The graph shows responses to the question: On average, how often do you interact socially with immigrants? Interaction means anything from a few minutes of conversation to doing an activity together. The interviews (face-to-face) were carried out in the period 2–29 November 2021. The EU27 data contain 26,510 interviews, and the Swedish data contain 1070 interviews.

Despite the positive attitude towards migrants, integration in terms of employment is lagging in Sweden (Statistics Sweden 2021). In fact, among all OECD countries, Sweden stands out with the highest unemployment gap between migrants and people borne in Sweden (Sandberg et al. 2022).

*1.3. Policy Shifts on Migration and Integration*

In Sweden, the political debate about migrant integration is associated with strong opinions about real and potential challenges. While some opinions are rooted in the 1930s, 1940s, or 1960s (Cholewo 2019), further ideas emerged in the wake of the in-migration of asylum seekers in 2015. Political sentiments about migration, as expressed in political narratives and campaigns, often build on the *perception* rather than on the actual number of migrants and are often associated with problems supposedly related to migration (such as crime) but often without robust evidence (Cholewo 2019). When conservative and nationalist circles (or other groups) tend to overestimate the number of migrants (Sides and Citrin 2007), this is often driven by misinformation or a particular political agenda (Fligstein et al. 2012). The 'sheer threat' of immigration may, therefore, not be reflected in real numbers.

Nevertheless, the in-migration of asylum seekers to Sweden in 2015 did put further pressure on society as regards national reception and integration facilities, and not least on housing, health care, schools, and other welfare services, especially in bigger cities. At the time, many EU governments shifted the responsibility for migration politics upwards to inter- and supranational levels and downwards to local municipalities while outsourcing to civil society many of the welfare services related to the reception and integration of asylum seekers (Caponio and Jones-Correa 2018). This policy shift was evident in Sweden, as were further changes in reception management and in-migration politics. In short, municipalities

and civil society organizations played a central role in filling the voids between authorities in the public sector and facilitating both the reception and early integration of asylum seekers (Stiernström et al. 2019).

Integration in Sweden is meant to benefit the whole society. Policies and regulations are designed to cover education and employment via general measures (Dahlstrom 2004). If social integration was previously seen as the prerequisite for entering the labor market, it is now the reverse—economic integration via the labor market is the suggested entry-point for social integration in Sweden. This is paralleled by a shift in the political responsibility for migrants from social authorities to employment agencies, thus representing a shift away from welfare institutions to the Swedish Public Employment Service (SPES). However, due to political pressure from liberal parties to privatize the SPES, more than every other of its agencies shut down in 2019, especially in small municipalities, and the staff was reduced by 35 percent (Wicklen 2019).

This policy shift is unhelpful when the number of employment seekers, including migrants, is growing, as is also the concern for their integration. Given the persistent unemployment gap between people born outside (18.4 percent unemployed) versus inside Sweden (4.4 percent) (SCB 2023a), the new policy to prioritize employment as a precondition for social integration may delay or reduce asylum seekers' opportunities to integrate into society. Integration must therefore be understood and addressed within the wider context of economic cycles, human capital, public opinion, and civil society. This calls for an analysis taking a broad political economy approach.

To sum up, the political economy in Sweden is characterized by institutional continuity but also decisive change. First, in terms of core features of the economy, the current labor market suffers more from unemployment than from the labor shortage typical of the 1950s–1960s. Second, and regarding status and region of origin, migrants are more often asylum seekers, mainly from the Middle East (Afghanistan, Iraq, Syria) and Africa (Horn of Africa, West Africa), rather than migrant labor from Scandinavia and Europe. Third, Sweden is a post-industrial society with a growing care and service sector (in need of labor) rather than a booming industrial society in need of workers, except for a growing demand for industrial labor in branches associated with a decarbonized economy, mainly located in the north of Sweden. Fourth, politics are less influenced by historical social democratic solidarity than by contemporary neoliberal thinking and anti-migration populism. Fifth, since its EU membership inclusion in 1995, Sweden has been expected to collaborate regionally and adhere to EU recommendations and regulations.

## 2. Theory about Society

Theoretically, we apply a sociological perspective to discuss the integration of asylum seekers/grantees with a particular focus on civil society organizations as facilitators. We proceed from Granovetter (1973, 1983) stating that 'weak ties' to acquaintances are more important for social change (integration in our case) than 'strong ties' to family and close friends. Accordingly, we propose that civil society can potentially provide weak ties that promote integration. Further, we draw on Fligstein and McAdam's theory on strategic action fields (Fligstein and McAdam 2011) to discuss the wider social order and political context within which civil society is operating and to understand the interplay between the political regime determining the national migration policy, municipalities receiving migrants, and civil society facilitating integration. Finally, we engage briefly with framing theory (Jerneck and Olsson 2011) to underline the importance of political communication and the powerful role of media in shaping attitudes about migration. Besides, we apply a few concepts on demographic dynamics and structures presented below.

### 2.1. Three Concepts: Demographic Opportunity, Cosmopolitanism, Inclusive Practices

Although international migration, especially of working-age groups and children, may help alleviate rural population decline, asylum seekers' ability to stimulate the economy in municipalities strained by out-migration is understudied (Hedlund et al. 2017). A

particular question, therefore, divides demographers, i.e., to what extent can the inflow counterbalance out-migration from rural areas in recipient countries? Some argue that international migration does revitalize rural communities (Hedberg and Haandrikman 2014; Woods 2016; Carson et al. 2017), while others disagree (Hedlund et al. 2017). For rural municipalities in Dalarna in mid-Sweden, scholars have shown that despite mounting pressure on education and housing, the influx of asylum seekers in 2015 was an energy boost to schools and an opportunity to vitalize both rural and civil society (Stiernström et al. 2019). Even if asylum seekers may later leave rural areas, we propose that they may serve as a *demographic opportunity* for dwindling rural communities. Whether that opportunity will materialize into long-term benefits will depend on political willingness in the municipality, the strength of the local economy, and successful integration.

To better conceptualize if and how asylum seekers/grantees succeed or do not to establish a livelihood in a rural community with little or no previous experience of migrants from the global south, Woods refers to 'rural' versus 'precarious' *cosmopolitanism* (Woods 2018). From the 1970s, the small Irish community of Ballyhaunis had a high influx of inhabitants from Asia and the Middle East. In the economic crisis of 2008, their comparatively successful establishment in this small remote community turned from 'rural' to 'precarious' cosmopolitanism due to economic contraction and lost job opportunities, not least for migrants (Woods 2018). A similar pattern was seen in the small Irish community of Gore, with a high share of migrants from Brazil (Woods 2018), and on the Norwegian island of Herøy (Aure et al. 2018).

Drawing on interviews about the reception of asylum seekers in Sweden exemplified by the city of Malmö, Fry and Islar argue that successful integration of asylum seekers should build on extended collaboration between civil society and municipalities and be based on *inclusive practices* rather than on the outsourcing of welfare services from the municipality to civil society (Fry and Islar 2021). In our study, we confirm that inclusive practices are important for integration, also in rural society.

*2.2. Three Strategic Action Fields: Political Regime, Municipality, and Civil Society*

To understand social change and continuity, it is helpful to speak of social order and strategic action fields (Fligstein and McAdam 2011) referring to how individuals, groups, and organizations interact in relation to structures and institutions (norms, rules, values). Social order can foster community, consensus, and belonging but also produce and reproduce oppression and resistance, resulting in tension or conflict. Interaction occurs within and between collaborating and competing strategic action fields inhabited by incumbents protecting the dominant social order and challengers resisting it. Incumbents repeat and institutionalize their practices to stabilize their field while challengers seek to disrupt it, transform it, or pave the way for competing strategic action fields. Apart from defining the fields and their actors, features, stakes, and discursive and material skills and tools, it is important to identify how strategic action fields react to external incidents and 'shocks' (Fligstein and McAdam 2011) like the 2015 in-migration taking governments by surprise.

Inspired by this theory, we identify three strategic fields: the national political regime; the recipient municipalities represented by politicians and officials; and civil society. The national political field is primarily driven by regular elections, the outcome of which determines the mandates for governing the economy and society at large, including how to regulate migration, asylum-seeking, labor market institutions, and various integration support systems. Similarly, the political mandate in municipalities is determined by local elections following campaigns, while officials act and perform based on rules and regulations set by politicians at multiple scales.

The composition and power of the national political regime and the political representation of municipalities are determined by rhetoric, narratives, and discursive tools that political parties use in national, regional, and municipal election campaigns followed by elections (every fourth year) and the political mandate resulting from that. In contrast, civil

society organizations are driven by benevolence and trust, and they design their activities in line with their respective ideology and mission while still being flexible. Primarily, this field is driven by members' everyday practices, initiatives, participation, and experiences (SOU 2017).

### 2.3. The Emergence of Populism as a Fourth Strategic Action Field

In response to the huge in-migration in 2015, the national political regime in Sweden decentralized and delegated the responsibility for asylum seekers' housing and schooling to municipalities. In turn, it outsourced an increasing share of its responsibility for various welfare services to civil society actors who also actively engaged in the immediate reception of asylum seekers. This quick and resolute response is documented in both urban (Fry and Islar 2021) and rural settings (Stiernström et al. 2019).

In 2015, when the pressure on housing, health services, and schools increased, civil society organizations admitted the stress while asserting that the situation was manageable (Stiernström et al. 2019). They maintained and even intensified their capacity to keep a consistent humane attitude towards, and practices for, successful integration. In parallel, populist views and voices increasingly challenged the incumbent national political regime on the generous migration policy (Ekman and Krzyżanowski 2021). This triggered a drastic change not only in the rhetoric around migration but also towards stricter migration policies with regulations and border control, resulting in a dramatic decline in asylum seekers (Figure 1) from November 2015 (SOU 2017). In national politics, the 2015 in-migration was soon framed as a problem and a crisis in response to the massive media coverage reporting on the shortcomings of authorities and the pressure on the welfare system (Cholewo 2019). But many authorities and municipalities who worked under increasing pressure did *not* report any collapse of routines or regulations (Stiernström et al. 2019).

The crisis rhetoric did not change attitudes in society more profoundly. However, given the spread and strength of populist anti-migration sentiments, we consider this to be a serious challenge and, thus, an emergent strategic action field. It is well organized with strong transnational ties, notably in France, Germany, Hungary, Italy, and the Nordic countries, and driven by anti-establishmentarianism (Bo' et al. 2023). In contrast, we see civil society as part of the 'generous incumbent regime' rejecting the anti-migration sentiments promoted by populism. Here we propose that populism as an emerging fourth field will succeed only if it also has the capacity to transform civil society.

## 3. Civil Society

Civil society can be defined as a space at the nexus of society, the economy, and the private sphere where citizens work together 'towards what they perceive as the common good' (Lang 2021). Importantly, civil society is 'an arena, separate from the state, the market and the individual household, where people, groups, and organizations act together for the common good and interest' (SOU 2017). Based on ideology or a particular mission, they seek to contribute to social development and problem-solving in relation to both people and municipalities. This echoes an old and widely held assumption spanning from Tocqueville, Tönnies, and Durkheim to Putnam that civil society participation is conducive to social trust. Even if empirical evidence is weak, this assumption seems valid at the aggregate level of nations (Newton 2001). While Putnam famously described a drastic decline in social capital in the USA since the 1950s, assessed through participation in civil society (Putnam 2000), Rothstein found no evidence for such a decline in Sweden, neither in trust nor in participation (Rothstein 2001). Recent research shows that Sweden and Norway stand out with high, steady, or even increasing participation in civil society organizations: around 50 percent of the population participate in at least one civil society association (SCB 2018; Qvist et al. 2019). Further, in the year before the 'asylum crisis', the total value of voluntary work in Sweden was estimated at 131 billion SEK, representing 3.32% of GDP (SCB 2018), an exceptionally high share in a European context.

Swedish civil society is well-anchored (Stiernström et al. 2019). It is characterized by a corporatist model wherein negotiations and joint decision-making occur in an institutionalized alliance with the state. Historically, civil society organizations were already crucial in building, expanding, and strengthening the welfare state from the late nineteenth century onwards (Lundberg 2020). In the post-war era 1950–1970, they facilitated the integration of migrant workers from Nordic countries, mainly Finland, and from Southern and Eastern Europe (Nilsson 2004). Again, in the 2015 'asylum crisis' in Europe, civil society played a crucial role in the reception and integration of asylum seekers in Sweden, as documented by authorities (SOU 2017) and scholars (Fry and Islar 2021). However, social relations are changing toward becoming more pluralistic and less institutionalized (Arvidson et al. 2018). In addition, economic support has shifted from unconditional grants to short-term contracts (Lundberg 2020). The role of civil society has also been characterized by increasing uncertainty regarding the division of responsibilities between different levels (state, region, municipality) and spheres (public, private, civil society), especially since a major reform in 2010 (Lidén et al. 2015). In line with neoliberal politics, certain subcontracted civil society organizations now act as alternative providers of welfare services (Reuter et al. 2012). Such outsourced publicly financed services are increasing in every welfare sector (Scaramuzzino and Scaramuzzino 2017).

Voluntary immigrant organizations are found in most countries and represent a special type of civil society organization worth mentioning (Babis 2014). Each of them draw their members from a particular country or region and specialize in assisting migrants in language training, administrative support, facilitating communication with their country of origin, religious services, and employment seeking. In Sweden, some 52 such organizations are active in various ways in supporting integration (Frödin et al. 2021; Fredholm et al. 2022; Sandberg et al. 2022). Theoretically, we would refer to them as providers of strong rather than weak ties, but empirically they are part of the civil society support system for migrants.

Importantly, policymakers pay more attention to civil society organizations as arenas for dealing with social challenges (Fundberg 2017) and promoting integration. In many municipalities, the local government is increasingly engaging in partnerships to support hobby clubs as a medium to foster communal values, social integration, and welfare provision (Ekholm 2017). Moreover, state funding has been directed specifically to actors providing sports activities and language education (Stiernström et al. 2019). In line with that, state funding to civil society organizations increased by Euro 20 million to compensate for all voluntary work in 2015 (SOU 2017).

Although hobby and sports clubs are praised as integration tools in the policy field (OECD 2016), scholars caution against idealization by pointing to issues of exclusion and racism (Fundberg 2017). Studies have shown that norms in hobby clubs may cause a sense of alienation among minorities (Spaaij 2015; Spaaij et al. 2015), especially if integration translates into assimilation, focusing on eliminating differences and making minorities integrate into the majority. Similarly, research on the role of hobby clubs in Denmark shows that there are situations where these can be criticized for reproducing established political goals (Agergaard et al. 2016) rather than opening for multiple voices.

This shift in civil society activities 'from voice to service' is mentioned widely in the literature (Scaramuzzino and Scaramuzzino 2017; Linde and Scaramuzzino 2018; Lundberg 2020), and scholars have expressed concerns regarding the weakened democratic role of civil society. When civil society increasingly provides public services, its role in advocacy and resistance runs the risk of becoming weakened or depleted (Lundberg 2020). Yet, research also shows that advocacy activities among civil society organizations continue to be strong, not the least concerning migration. The Church of Sweden is engaged in issues associated with asylum rights and can use its institutional role and resources to engage in Swedish politics (Linde and Scaramuzzino 2018). In a survey of 2791 Swedish civil society organizations, researchers found a deep-seated advocacy culture (Arvidson et al. 2018) exemplified by hobby clubs establishing relations with municipal decision-makers as an

advocacy strategy (Ekholm 2019). Further, scholars (Arvidson et al. 2018) conclude that recent changes in funding arrangements and expectations of civil society organizations as increasing 'service providers' do *not* put the culture of advocacy at risk. Instead, advocacy remains a cornerstone of the Swedish model (Arvidson et al. 2018).

To sum up, civil society organizations unite on the national level through federations and umbrella organizations. The municipal level constitutes the main arena for their activities and mobilization of resources, whereas the European level adds a complementary layer of opportunities in multilevel governance (Scaramuzzino and Wennerhag 2019). Nevertheless, the development towards a pluralistic political system is influenced by EU membership and reflects a Europeanization of Swedish civil society.

## 4. Methodology

This study is explorative, striving to be explanatory. In a mixed methods approach, we combine a qualitative content analysis on migration in local-regional media (See Supplementary S4) with a quantitative survey on reception, integration, and attitudes towards migrants sent to civil society organizations (See Supplementary S1 and S2).

Based on our pre-understanding of national and regional migration politics and Swedish demography, we identified three types of rural municipalities (dwindling, dynamic, dissenting) with potentially different interests, resources, and capacities to integrate asylum seekers. In these geographically dispersed and purposively selected municipalities, we study how attitudes towards, initiatives on, and inclusive practices to support integration correlate with the demographic profile, political majority, and socio-economic structures. To deepen our understanding of public opinion and sentiments towards migration and migrants in various settings, we conducted a content analysis of local media reports. In doing so, we are particularly interested in demographic structure, employment opportunities, and if and how the profile of the local political regime influences the interest and ability of civil society to engage with migrants. For the media analysis, we selected one unit of analysis for each type: one dwindling in the north of Sweden (Åsele), one dynamic in the central south (Gnosjö), and one dissenting in the very south (Sjöbo). Below, we describe the types and their corresponding units.

*Dwindling, Dynamic, and Dissenting Municipalities*

Type 1: Small dwindling municipalities in need of a growing population to keep up private and public services. We assume that these municipalities seek demographic opportunities and are willing to receive any migrants. They are mainly located in the sparsely populated north of Sweden, dominated by an economy of small-scale agriculture and forestry, here exemplified by Åsele in Jämtland County. They have undergone a steady demographic decline since the 1980s and generally have a low percentage of foreign-born inhabitants (14%). Since the 1950s, Jämtland County has been best described by out-migration until a recent trend reversal thanks to international migration.

Type 2: Small dynamic municipalities in need of labor. We assume these municipalities adhere to rural cosmopolitanism and are willing to receive migrants, especially skilled workers. They are characterized by an entrepreneurial spirit and small-scale industries in need of labor. They are mainly located in sparsely populated areas in southern Sweden, here exemplified by Gnosjö, in Småland County. The population has increased steadily since the 1980s and has a rather high percentage of inhabitants born outside Sweden (34%), mainly in the Middle East.

Type 3: Small and well-to-do municipalities dominated by anti-immigration politics. We assume that they are unwilling to receive migrants and therefore call them dissenting municipalities. Yet, to balance such anti-migration sentiments, we assume that there may be counteracting forces that promote *inclusive practices*. They are mainly located in comparatively densely populated southern Sweden, here exemplified by Sjöbo, in Skåne County, receiving migrants from many parts of the world. Such municipalities have a

history of strong resistance against immigration and a low percentage of inhabitants born outside Sweden (13%).

In line with this typology, we started with a content analysis of news media articles from local-regional newspapers covering the three selected municipalities (Åsele, Gnosjö, Sjöbo) in 2009–2019. The objective was to screen attitudes and how local politicians and populations frame immigration, integration, and civil society. For that, we systematically analyzed how these issues are discussed in 'debate articles by local politicians', 'editorials', 'letters from the local population', and 'news reporting articles'. In the media analysis, we asked the following question:

- How do local politicians and populations frame immigration, integration, and civil society activities relating to (and possibly supporting) these issues?

Guided by the insights from our media analysis, we sent out a survey to civil society organizations in the three municipalities, later followed by a wider distribution to include civil society organizations in all 290 municipalities in Sweden. The objective here was to learn more specifically '*if and how civil society organizations engage*' in various activities to promote integration. In the survey, we asked:

- How is civil society active and important in integrating asylum seekers, and what kind of civil society actors are most or least engaged?
- How does civil society engagement vary with the demographic, political, and socio-economic conditions of the municipality?

As a starting point for analyzing the survey results, we categorized all 290 municipalities into our three types and analyzed the responses from civil society organizations accordingly. Our categorization is specified by the following quantitative and qualitative criteria:

- Dwindling: small municipalities (<20th percentile) with a declining population from 2010 to 2020 and with comparatively *high* unemployment (>40th percentile);
- Dynamic: small to medium-sized municipalities (<60th percentile) with comparatively low unemployment (<40th percentile);
- Dissenting: small to medium-sized municipalities (<60th percentile) with a *strong* anti-immigration political party (SD >90th percentile of the votes).

After the survey, and to identify more profoundly how civil society interacts with migrants, we supplemented our database. On demography, we added data on the percentage of inhabitants born outside Sweden, including changes in these numbers since 2010. Concerning politics, we added percentage votes in the 2018 election for all political parties. Concerning the socio-economic dimension, we added data on the percentage of unemployed, divided into three categories: Swedish-born, foreign-born, and total.

Guided by insight from the media analysis and survey, we asked two related questions:

- Why is civil society, in terms of attitudes and activities expressed in actual practice and in the responses to the EU 27 in 2021 (Figures 2 and 3), much more positive towards migrants than Sweden's restrictive migration policy?
- Why is there a discrepancy between national restrictive migration politics and municipally based civil society activities and attitudes towards migrants?

## 5. Materials

### 5.1. Media Analysis

To identify how ideas and perceptions about immigration vary between actors in the period of 2009–2019, we analyzed a total of 386 articles (Supplementary S4) on immigration from local newspapers in three municipalities: Åsele (65), Gnosjö (45), and Sjöbo (276). The newspapers were accessed in pdf-format from the database BIBSAM through Lund University. We used the NVivo software for the extraction and analysis of the relevant texts. Eleven codes were used that we had either predetermined as sensitizing concepts for the search or classified as reoccurring themes in the analysis: 'business', 'civil society other than sports', 'civil society sports', 'criminality', 'culture', 'demography', 'economy',

'employment', 'health', 'housing', and 'school'. For every code, we categorized texts according to type, such as 'debate article by local politicians', 'editorial', 'letter from the local population', and 'news reporting article'.

*5.2. Survey of Civil Society Organizations*

We developed a web-based survey that asked 72 questions, 21 of which were follow-up questions made visible when respondents chose certain answers (Supplementary S1). Migrants were people who have been granted residence permits in the last two years (i.e., since 2017). It started with questions aimed at identifying organizational characteristics followed by general questions on migrants' engagement and involvement in the organization, such as whether migrants are currently engaged in it and how, whether parents to migrants are involved, and general attitudes towards involving migrants in the organization. It then turned to practices and strategies that organizations used to reach out to migrants. After that, the survey posed questions about useful skills and knowledge gained from both sides from migrants' involvement as well as experienced or potential obstacles to including migrants. Here we asked whether members have increased their knowledge and understanding of different experiences and living conditions. We then moved on to questions on the organization's relationship with the municipality, i.e., the elected regime and its officials, and whether involving migrants (including asylum seekers and undocumented migrants) has affected the relationship. Lastly, we asked about the effects on activities due to the COVID-19 pandemic.

In Sweden, civil society organizations can receive municipal funding and are subsequently listed in a municipal register with contact information. We sent the survey via email to organizations in all 290 municipalities in Sweden. In each municipality, we selected organizations that could be expected to engage with migrants in business, culture, education, sports, and outdoor recreation or who have an ethnic, religious, or humanitarian profile or any other relevant community orientation. We reached out to 12,774 organizations and received 1338 responses, thus a response rate of just over 10 percent. The survey was open for one month. The actual response rate is probably higher than 10 percent because some contacted civil society organizations may no longer exist or had changed the contact person at the time of the survey. Respondents were anonymous, but the survey offered the option of providing contact details for follow-up interviews. The full breakdown in groups is shown in Supplementary S2, but in short, sports clubs clearly dominated the responses (*n* = 655, 49%), followed by cultural (*n* = 137, 10%), religious (*n* = 118, 9%), and outdoor organizations (*n* = 100, 7.5%).

For the analysis, we downloaded data from the survey into SPSS files. We generated descriptive statistics using frequency tables and cross-tabulations (with Chi2 tests) to depict general trends and themes in respondent data while considering highlights from the media analysis (Supplementary S3, Tables S1–S3). We corroborated the survey data with three selected dimensions of municipality data:

- Demographic dimension: Total population in 2018, percentage change of population from 2010 to 2020, percentage born outside Sweden in 2010, percentage born outside Sweden in 2018, *change* from 2010 to 2018 in percentage population born outside Sweden;
- Socio-economic dimension: Unemployment of inhabitants born in Sweden, unemployment of inhabitants born outside Sweden, total unemployment, difference in unemployment between inhabitants born in Sweden and outside;
- Political dimension: Results from a municipal election in 2018. Mainly the percentage votes for the Sweden Democratic (SD) Party which has the most outspoken anti-immigration program of the eight main political parties in Sweden.

We then used the three dimensions to categorize municipalities into the three types of dwindling, dynamic, and dissenting.

Finally, we tested for differences in responses between three categories of municipalities based on size: small (<50,000 inhabitants), medium-sized (50,000 to 100,000), and major

cities (>100,000). The survey questions are listed in Supplementary S1. In Supplementary S2, we give a full breakdown of the number of responses from different types of civil society organizations and a formal definition of the three types of municipalities. The statistical details of our analyses are found in Supplementary S3, Tables S1–S3.

## 6. The News-Media Analysis

There was a significantly higher number of newspaper articles on (dissenting) Sjöbo indicating that migration is a contested issue there and high on the political agenda. In all three cases, the economic aspect is debated the most. The code 'economy' is most frequent in Sjöbo (25%), followed by Åsele (18%) and Gnosjö (8%). Among local politicians, there are strong ambiguities about the economic effects of migration reception. As regards negative sentiments, both local politicians (from the Sweden Democrats) and the local population point to increasing tax expenditures resulting from the influx of migrants with a general understanding that the costs for receiving migrants are too high. In contrast, local politicians who oppose the high-cost argument claim that the integration processes should be improved and supported since every new inhabitant is a potential new taxpayer adding revenue to the municipality.

We see a clear difference in how economic implications are discussed in the three municipalities. Economy articles in (dwindling) Åsele are mainly positive debate articles and letters. The arrival of migrants is described as a "vitamin injection", as something to embrace, a possibility for growth rather than a burden, and a much-needed addition to the workforce. Articles from both Åsele and (dynamic) Gnosjö discuss forms of employment training programs and their positive effect on integrating migrants into the local industry and workforce.

Regarding education, Sjöbo articles mention various school forms for adult migrants that may more easily enable integration into the labor market. Investing in this is important since the migrants will become an important future resource for the municipality. At the same time, there are negative sentiments around employment. In news coverage interviews, the local population expresses a sense of perceived unfairness that migrants 'steal the jobs' from local inhabitants. Similarly, negative sentiments relating to housing are reflected in an editorial about house owners who oppose the construction in their neighborhood of accommodation for unaccompanied migrant children. In Åsele and Gnosjö, limited access to housing is seen rather as hampering the reception of asylum seekers.

Some articles about Sjöbo mention initiatives to combat prejudices and racism in schools. This suggests that while Sjöbo, in several ways, is a (seemingly) hostile environment for migrants, there are counteracting forces. The media analysis suggests that civil society in the Sjöbo region holds an anti-racist position and that several initiatives are explicitly working against xenophobia and hate crimes. Similar articles are not found in the other municipalities, which we interpret as a sign of an overall positive attitude towards migrants in those. Yet, none of the articles coded as 'civil society' are classified as debate articles, suggesting that local decision-makers may lack interest in using civil society as an arena to improve integration.

Civil society is mentioned more often in (dynamic) Gnosjö articles (43%) than in those on Sjöbo (10%) or Åsele (4%). The media analysis also suggests that Gnosjö has a strong civil society, both within sports and other areas. In contrast to Sjöbo, newspaper articles from Gnosjö frequently point out civil society as important for the integration process. In interviews and debate articles, local politicians attributed the successful integration of migrants to civil society actors who collaborated with the municipality. Here, civil society initiatives range from sports activities to language cafés, dance classes, and friendship matchings.

Articles from (dwindling) Åsele do not mention civil society initiatives much. None was coded as 'civil society other than sports', and only one article was coded as 'civil society sports'. When it is mentioned, the sentiment is positive and emphasized as a good way to integrate newly arrived children. This might suggest that while there is a generally positive attitude towards the reception and inclusion of migrants through civil society, the

demographic decline means limited resources for civil society to mobilize and perform welcoming initiatives.

Local newspapers in (dwindling) Åsele state that the arrival of migrants is positive for the local demography. According to local politicians and the local population, and regularly expressed in debate articles, the influx of migrants offers a solution to the municipality's sharp demographic decline, thus a demographic opportunity. All six articles coded with 'demography' articulate the positive effects of in-migration. News coverage articles present statistics on the population trend, and editorial letters, along with letters from the local population, express generally positive sentiments.

In (dynamic) Gnosjö, letters from the local population point in the same direction as in Åsele, stating that the arrival of migrants is positive and an opportunity for demographic and economic growth, thus a sign of rural cosmopolitanism. Representatives of the local industry point out in interviews that the influx of migrants is necessary for the survival of the industry. In (dissenting) Sjöbo, two news coverage articles mention the positive demographic effect attributing the growth of some rural municipalities in Skåne County to the arrival of migrants.

*Three Key Media Insights*

Three key insights on migration and integration emerge from the media analysis, one for each type of municipality—dwindling, dynamic, and dissenting. We interpret the insights based on the theory and concepts we presented above.

Åsele, the small municipality with a dwindling population in the sparsely populated north, is unanimously positive to in-migration and asylum seekers but seems to lack resources to mobilize impactful initiatives to seize the full demographic opportunity that international migration offers. Yet, civil society is still an important actor that can potentially offer the weak ties that may foster integration.

Gnosjö, the small dynamic and entrepreneurial municipality in a sparsely populated area in the south, is positive to in-migration (especially skilled labor). The community celebrates civil society initiatives and attributes successful integration to civil society actors and their activities, thus, like above, a sign of the strength of weak ties. But dissent is brewing, perhaps more in speaking than acting, and can probably be attributed to a slowly increasing unemployment rate. Hence, this situation may trigger a shift from 'rural' to 'precarious' cosmopolitanism (Woods 2018).

Sjöbo, the small dissenting municipality situated in the rural and comparatively rich agricultural region in the south, has a strongly negative attitude to asylum seekers—but civil society is trying hard to challenge the dominant view of the political establishment. One key task for civil society is to oppose the hostile environment towards migrants by explicitly addressing racism and xenophobia and promoting counteractive initiatives and inclusive practices. Again, a sign of how the weak ties between civil society organizations and social groups can be used to strengthen the support for migrants and, by its extension, their integration into the municipality.

## 7. The Survey of Civil Society Organizations

Overall and regardless of geographic location or type of municipality, we found that civil society organizations in Sweden have a very positive attitude towards migrants [*What is the attitude towards the participation of migrants in your organization? 97% positive or very positive, 1% neutral, and 2% negative*]. Of the organizations with the active participation of migrants, they are seen as an asset by almost 80% [*Migrants from different cultures are an asset for your association: Agree 79%, Neutral 18%, Disagree 2%.*]. Organizations also recognize the importance of participation in civil society for the integration process [*Association life, in general, is important for the integration of migrants: Agree 81%, Neutral 16%, Disagree 3%.*] (Table S2).

Regarding participation, we found that, on average, 38% of all organizations reported that migrants are currently active in their association. We found a significantly ($p = 0.011$)

higher participation in the large cities (46%) compared to small (34%) and medium-sized (37%) ones (Table S2). Among rural municipalities, the participation was significantly ($p < 0.001$) higher in dwindling (33%) municipalities than in the dynamic (25%) and dissenting (27%) ones (Table S1).

On interaction with the politicians and civil servants in the municipality, we found that civil society organizations report that they have a good or very good relationship: [*How would you describe the association's relationship to the municipality (its politicians and officials)? Positive 67%, Neutral 23%, Negative 10%*]. There was a significant ($p < 0.001$) difference in responses based on the size of municipalities, where small (71%) and medium-sized (72%) municipalities more often reported a good or very good relationship compared to major cities (54%). The financial support to civil society organizations for integration activities did not vary significantly across small municipalities to major cities (Table S2).

If we break down the responses based on faith-based (of which a majority are Christian) and non-faith-based organizations, we find several significant differences regarding their attitudes and engagement with migrants (Table S3). The analysis shows that faith-based organizations have significantly more favorable attitudes toward migrants than non-faith-based ones (93% versus 85% positive attitudes). They also engage migrants (53%) and their parents (49%) much more often than non-faith-based organizations (36% and 20%, respectively). In addition, faith-based organizations are much more active in recruiting migrants and parents of migrants and receive financial support for this more often (27%) compared to non-faith-based organizations (21%) (Table S3). The positive attitude held by Christian organizations confirms arguments about the Church of Sweden in previous studies (Ideström and Linde 2019).

Getting a job is important but often difficult for first-generation migrants (Gorodzeisky and Semyonov 2017). Employment can therefore be seen as an important indicator of integration. However, there is no significant relationship between levels of unemployment in the municipality and organizations' attitudes towards migrants, nor with their levels of participation in civil society. We tested both for employment in general and for the difference in unemployment between people born in and people born outside of Sweden.

There are some highly significant differences between the three types of rural municipalities. In dwindling municipalities, civil society organizations engage migrants, and their parents, much more frequently than in the other municipalities, particularly compared with the dissenting ones.

Attitudes towards migrants show some differences between the three types, even though the significance is low ($p = 0.13$). Civil society organizations in dwindling municipalities have more positive attitudes (96%) than those in either dynamic (86%) or dissenting (84%) municipalities. In dwindling municipalities, civil society organizations are also more actively working to recruit migrants (33%) than in dynamic (21%) or dissenting (13%) municipalities. The financial support given to civil society organizations for integration activities does not vary significantly across the three types of rural municipalities. Civil society organizations in the three types of municipalities responded differently to the question: [*Do you feel that the municipality's goals for integration and your goals for integration agree?*]. Due to few responses, we could not see if there was any significant difference, but the response (YES) ranged from 58% in dwindling to 31% in dissenting municipalities. (Table S1).

*Three Key Survey Insights*

Three main insights emerge from the survey: one referring to attitudes, another to integration, and the third to interaction. First, as regards attitudes, it is clear that, at large, civil society organizations have an overwhelmingly positive or even very positive attitude towards asylum seekers and their participation in organized activities. This is confirmed by and reflected in the EU27 barometer (Figures 2 and 3). Previous research has also shown that emerging social relations between civil society and asylum seekers facilitate integration (Stiernström et al. 2019).

Second, as regards the integration of asylum seekers into Swedish society, organizations trust that civil society itself has the capacity, skills, and discursive tools to contribute decisively to facilitate that process, and that they also should do so. This is particularly clear among faith-based Christian organizations, as seen in other studies (Linde and Scaramuzzino 2018). Moreover, it echoes the government's effort to decentralize responsibility to civil society organizations by giving them the mandate to provide certain welfare services. It is evident that civil society seeks to live up to this. A profound study on rural municipalities in Dalarna, Sweden, shows that civil society was central in assisting the public sector and creating social relations with asylum seekers (Stiernström et al. 2019). That study identified three types of civil society units engaged in the reception and integration of asylum seekers: new ad hoc groups emerging as a spontaneous response, established organizations with previous experience from working with asylum seekers, and hobby clubs attracting and recruiting members (Stiernström et al. 2019). While some of these organizations were involved in the reception and immediate needs of asylum seekers (clothing, toys, transport, translation) or in serving as a bridge to the municipality, others, such as sports clubs, activated new members (Stiernström et al. 2019).

Third, as regards interaction between strategic action fields, the dynamic between civil society organizations and the municipality is working well, according to respondents, and it is characterized by a division of labor and mutual trust rather than competition or conflict. The level of financial support from municipalities does not vary between types of municipalities, not even in the dissenting ones.

To conclude here: With increasing financial means, civil society organizations would have the potential to improve their material resources and tools even further. In the results from the survey sent to civil society organizations, there are no signs of or any support for the dramatic shift of asylum politics that emerged and was implemented in late 2015 in the aftermath of the wave of asylum seekers. Civil society organizations responding to the survey did not report any support for anti-immigration sentiments or policies despite increasing municipal representation of political parties with anti-immigration policies in the 2018 election. Nor did the profound study on Dalarna identify any anti-migration sentiments (Stiernström et al. 2019). It would be interesting to initiate a follow-up study on this for the 2022 election, where populist parties increased their mandate even further and at all scales.

## 8. Discussion and Conclusions

Civil society in Europe is central to democracy and legitimacy in the EU (Lang 2021). This is evident in Sweden. The division of labor between the Swedish state, municipalities, and independent public agencies requires a strong capacity and competence to network, cooperate, and coordinate activities in times of social change. It has been confirmed that civil society organizations, together with individual volunteers, developed an impressive collective ability to assist asylum seekers and government agencies in the acute phase of in-migration in 2015 and the subsequent integration (SOU 2017). With that, it became evident that civil society organizations are important, even necessary, in the reception of migrants (SOU 2017). After 2015, many municipalities continued their collaboration with civil society organizations and considered them to be equal and vital partners in community work (SOU 2017).

In 'difficult times' we may therefore expect that civil society organizations will again reinforce their agency to facilitate social integration, as suggested by the OECD in 2016 (OECD 2016) and reiterated by the EU commission in 2020 (European Commission 2020). As a foundation for that, Swedish authorities have declared that migration is part and parcel of global society and that a rapidly increasing migration flow is not a threat to society or the welfare system in and of itself, even if the situation may become very strained (SOU 2017). This is also confirmed in research.

In line with that, we have seen that civil society organizations express overwhelmingly positive attitudes toward migrants. While we found that civil society organizations of

varying profiles have a positive or even very positive attitude towards as well as active engagement with migrants, it is a further observation that faith-based organizations (particularly Christian ones) stand out as even more engaged than non-faith-based organizations in the aftermath of 2015. From that, we now confirm that 'weak ties' provided by acquaintances (such as civil society) play a decisive role in social change (Granovetter 1973, 1983) in terms of paving the way for employment, integration, and mobility among asylum seekers. Admittingly, strong ties also play a positive role.

Despite such strong civil society engagement in the integration process and despite the positive attitude towards migration and migrants among people in Sweden as reflected in the EU barometer 2021, it is notable that Sweden shifted towards strict migration politics and restrictive asylum regulation, starting at the end of 2015. The remaining question is, therefore: why did Sweden make a drastic shift to restrictive migration politics, and how come civil society organizations challenge this?

According to one visible observation, the massive media attention to the 'asylum crisis' served as a trigger for the shift. Another, and politically more precarious reason, was the immediate threat of a government crisis pushed forward by the populist party that wanted an end to the influx of asylum seekers, to which the government responded forcefully by re-introducing border controls (SOU 2017). A further and more profound observation points to how established political parties in the EU had not responded in time to a growing discontent concerning the crumbling welfare system, creating a political opportunity for right-wing populist parties to gain political influence based on the rhetoric of a faltering welfare state. In some countries, populists successfully created new parties. In other countries, they were able to coopt existing parties and gain electorate support by skillfully using the discursive tool of tying the welfare issue to migration, thereby, step by step, making asylum seekers into scapegoats and a threat to the whole system.

Since 2015, the electorate's support for populist parties propagating anti-migration policies has been growing. General elections in Sweden in 2018 and 2022 resulted in a substantial increase in the number of municipalities with a strong representation of the anti-immigration party Sweden Democrats (SD). From their low influence before 2018 and almost negligible position before 2014, the party became the second largest party in the national election in 2022. Since then, it has been represented in the governing majority of 34 out of 290 municipalities, holding the chair in ten of those.

In seeking an answer to how this increasing political resistance to migration impacted civil society organizations, we tested for a relationship between SD representation and variables related to attitude towards and involvement of migrants, both in a local news media analysis and a survey of civil society organizations. However, none of the variables differed significantly between municipalities with high or low SD representation. Therefore, we conclude that civil society organizations keep a positive attitude towards migrants irrespective of the political support for SD in their municipality. Hence, we do not see many signs of how the emerging populist regime, i.e., the challenger of the incumbent regime, is capable of penetrating civil society organizations. Below, we seek to explain the discrepancy in attitudes towards migrants between civil society and the anti-migration policy reflected in the elective vote.

Speaking of the core social science concepts of structure and agency, it is evident that multiple interacting agents use a range of resources and skills to act 'with and against others' in the context of structures and institutions (Sewell 2004). In doing so, agents are empowered by access to discursive and material resources, and their agency builds on the capacity to interpret and mobilize these resources in a variety of ways. The agency also varies with incentives and motives and depends on the capacity to draw on complex repertoires to engage creatively in interaction (Sewell 2004).

The increasing elective support in Sweden for political parties with anti-immigration policies on their agenda can be seen as a discursive structure or an emerging strategic action field not yet fully consolidated. It is not stable enough to withstand the challenges posed by civil society organizations that use their own agency to explicitly express willingness

to engage with migrants in support of integration and who do so via inclusive practices. There are several intertwined reasons for this.

First, civil society organizations tend to be driven by benevolent principles to serve a particular ideology, practice, or social goal. They also tend to advocate social interaction and belonging, like that in faith communities and hobby clubs, what the sociologist Ferdinand Tönnies would call *Gemeinschaft* (Waters 2016).

Second, it is obvious that civil society cannot (and should not) compensate for all the municipal and regional services that are needed and expected to be provided by state authorities at multiple scales. Services may be missing or in short supply in many municipalities, especially in dwindling ones, be it in health care, schools, transport, or communication. Hence, rural municipalities have reasons to attract migrants and seem to engage more strongly than big or bigger cities to support civil society organizations. This is probably grounded in the incentive and intention to grasp the demographic opportunity, meaning that an increasing migrant population of working age or younger can help a municipality to keep or expand its welfare services through work and taxes.

By drawing on the theory of strategic action fields, we can conceptualize and better illustrate the relationships between the three active fields in this debate: the national political regime, the municipality with its officials, and civil society organizations. By discussing their roles and interaction, we will get closer to explaining the discrepancy between the anti-migration politics seen as a discursive structure and the practiced agency (practice) of supporting the reception and integration of asylum seekers. One way to do that is to conceptualize populist anti-migration politics as a severe challenge to the incumbent regime of generous migration politics and, thus, an emerging strategic action field.

Political parties create narratives based on three main components: party-specific ideology, particular signifiers, and symbolic reasoning (aimed at) appealing to voters, thus offering a partial and ideologically biased description of reality. Narratives are presented by party representatives and diffused through a range of communication channels and political campaigns. Political constituencies respond to the narratives in their voting behavior once every four years. It should be noted that political parties can change rules, regulations, and financial conditions but have a very limited mandate to change how civil society organizations operate. Apart from the time of elections, most individuals are passive in relation to municipality politics, while many (even every other person in Sweden) are engaged in civil society. While political parties rely on votes only, once every four years, civil society organizations consolidate their existence based on participation and by attracting individuals, including migrants, to engage actively, regularly, and continually in all sorts of activities and practices.

To conclude, we argue that populism has limited persuasive power in effectively spreading anti-migration sentiments and their anti-establishment program when civil society, as a profoundly grounded yet flexible institution, acts as a counterweight by doing good for society.

**Supplementary Materials:** The following supporting information can be downloaded at: https://www.mdpi.com/article/10.3390/socsci12070403/s1, Supplementary S1: Survey questions to civil society representatives in all 290 municipalities; Supplementary S2: Descriptive statistics of the survey and formal definition of the three types of municipalities (dwindling, dynamic, dissenting); Supplementary S3: Cross tabulations of the survey results (Tables S1–S3); Supplementary S4: Descriptive statistics of the media analysis.

**Author Contributions:** Conceptualization, L.O.; methodology, L.O. and C.F.; formal analysis, C.F. and A.B.H.; writing—original draft preparation, L.O. and A.J.; writing—review and editing, L.O. and A.J.; project administration, L.O.; funding acquisition, L.O. All authors have read and agreed to the published version of the manuscript.

**Funding:** This research was funded by the European Commission's Horizon 2020 Research and Innovation Programme. Project: Migration Governance and asylum crises (MAGYC), Grant agreement number 822806.

**Institutional Review Board Statement:** Ethical review and approval were waived for this study because no questions of personally sensitive issues were involved (LUCSUS 9 April 2020). All questions of the survey refer to civil society organizations and not to respondents.

**Informed Consent Statement:** Not applicable.

**Data Availability Statement:** Data can be made available upon request.

**Conflicts of Interest:** The authors declare no conflict of interest.

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
