# Peer review of "Civil Society and Social Integration of Asylum Seekers: The ‘Strength of Weak Ties’ and the Dynamics of ‘Strategic Action Fields’"

_socsci, doi:10.3390/socsci12070403_

Round 1

Reviewer 1 Report

A very useful article in my opinion and I really enjoyed reading it. Please find some minor revisions below:

Article is cluttered with way too much information, that can be presented in a condensed/succinct manner. The introduction needs to give a clear overview of the article aims and structure. It will be useful if methodology comes earlier on. I understand the inclusion of media analysis, however, it needs to be strongly justified in further few sentences. There is a need for a separate conclusion. Finally, kindly re-think the inclusion of three diagrams/graphs, as it is very distractive. 

Overall, a very interesting article and a useful contribution to the literature. With some enhancements, the piece will become stronger and punchier.

This is fine.

Reviewer 2 Report

An interesting and well written article. I only have a few comments intended to strengthen what is already a very strong submission.

Your empirical study of the perception of migrants in Sweden is based on the discoursive analysis of local news media in three communities and a survey questionnaire that was sent out to all civil society organization in Sweden. Unless I missed it I could not find Tables A1, A2, or A3 that are referred to in your text. (pages 12 and 13) 

With respect to your analysis of the local news media, it would have been helpful to provide further quantitative details regarding your examinations of the print media articles that you selected for your study. How many articles were selected from each of the three local print media? How were they categorized in terms of the type of articles; e.g., news, sports, editorials, etc. Some of this is provided in the last section of the paper titled, "8. Materials and Methods," that appears to be an Appendix to your submission. But this appears to be supplementary to your analysis rather than part of it. For example, how many newspapers in total were drawn upon? Why were the newspaper articles overwhelmingly from only one of the three municipalities and how did this impact on your individual and general findings and overall conclusions? My point here is that you need to provide a quantitative breakdown of the articles and then present your findings not merely report them to the reader.

For the survey of civil society organizations there are references to Appendix 1 and 2. But, again, I was unable to find this in your submission. Appendix 1 was your survey instrument of 72 questions. It was sent out to 12,744 organizations and you had 1,338 responses. This is impressive. What is the breakdown of these civil society organizations by type; for instance, religious, social services organizations, legal clinics, etc? Did you obtain ethics review approval to conduct this study with these human participants? If so, you should indicate this in your text. There is a reference to the possibility of follow-up interviews with the respondents. Was this done? Appendix 2 includes, although this is not entirely clear, the Tables of statistical details for the media analysis, but, this seems to be missing.

While I found the last substantive section of the article very interesting, "7. Discussion and Conclusions," my sense was that it went well beyond the empirical findings of the local media analysis and the survey results and undertook a further analysis and discussion of the support for anti-migrant/refugee sentiments and politics in Sweden to try to explain the shift of government policies following the so-called "refugee crisis" in 2015. While very interesting, I do not think it was fully or sufficiently grounded on the findings of your two empirical studies. More thought needs to be put into how to connect the two empirical studies to the electoral politics and public policies shifts as a consequence of refugee flows to Sweden.

Please note the following that needs your attention:

At page 2, second paragraph, please note it is UNHCR and not UNCHR.

At page 13, first paragraph, last sentence, it is not "borne" but "born."

Reviewer 3 Report

The topic is interesting, but the article would need major revision before being published. Even though the paper deals with the 2015 refugee in-flow, a reference to the latest migration flows would be good. In general, the text is very redundant and very long in the conceptual and historical part, while the empirical part stays extremely superficial. Therefore, it would be recommended to shorten the first parts and to go more in detail in the empirical part. Here it would be also good to rethink if the results of all different methods would be needed - as each of them are only tackled shortly, or rather go more in-depth with one of the methods. Moreover, having a chapter 4 methodology and then having a chapter 8 "materials and methods" is quite difficult to follow. A combination of both methodological chapters in chapter 4 would be recommended.

Round 2

Reviewer 2 Report

Thank you for the revisions. Just a couple of concerns, despite accepting your submission for publication in its revised form.

-- Your explanation for not getting research ethics approval is not persuasive. You sent the questionnaires to individuals in the civil society organizations and got them to complete the forms. Your point about sending them to institutions does not have the ring of truth.

-- I sense that something is missing from your analysis and conclusions. It is not surprising to find civil service organizations being positive on migrants, especially, faith based groups and sports clubs, who service migrants and are integral to the newcomer's integration process and the viability of the organizations' congregations/membership. The findings with the media are interesting as well but it might be due to the nature of the media outlets you selected. Are they liberal newspapers as opposed to conservative newspapers?

-- If civil society groups and the media are positive to immigrants and asylum seekers then why are anti-immigrant political parties gaining support? Clearly, something else is going on here that you have been unable to detect from the data you have collected. Shouldn't you acknowledge as much?

-- Trying to explain the paradox of swings in electoral support from civil society organizations and limited media coverage analysis on migration concerns located in small largely rural communities that favour in-migration seems far too constrained and narrow when what is typically undertaken to explain such electoral shifts and seeming anomalies are studying opinion polling, electoral campaigns, voting analysis, campaign financing, etc. 

Several suggestions on wording:

Line 8 - you over use "profoundly." Suggest you find a suitable alternative.

Line 755 - you use "electorate" when "electoral" is probably better.

Reviewer 3 Report

The manuscript has improved a lot.